# LoSA: Locality Aware Sparse Attention in Diffusion Language Models

Haocheng Xi [1]   Harman Singh [1]   Yuezhou Hu [1]   Coleman Hooper [1]   Rishabh Tiwari [1]   Aditya Tomar [1]
Minjae Lee [2]   Wonjun Kang [2]   Michael Mahoney [1,3,4]   Chenfeng Xu [5]   Kurt Keutzer [1]   Amir Gholami [1]

## Abstract

Block-wise diffusion language models (DLMs) generate multiple tokens in any order, offering a promising alternative to the autoregressive decoding pipeline. However, they still remain bottlenecked by memory-bound attention in long-context scenarios. Naïve sparse attention fails on DLMs due to a KV Inflation problem, where different queries select different prefix positions, making the union of accessed KV pages large. To address this, we observe that between consecutive denoising steps, only a small fraction of active tokens exhibit significant hidden-state changes, while the majority of stable tokens remain nearly constant. Based on this insight, we propose LoSA (**Lo**cality-aware **S**parse **A**ttention), which reuses cached prefix-attention results for stable tokens and applies sparse attention only to active tokens. This substantially shrinks the number of KV indices that must be loaded, yielding both higher speedup and higher accuracy. Across multiple block-wise DLMs and benchmarks, LoSA preserves near-dense accuracy while significantly improving efficiency, achieving up to $+9$ points in average accuracy at aggressive sparsity levels while maintaining $1.54\times$ lower attention density. It also achieves up to $4.14\times$ attention speedup on RTX A6000 GPUs, demonstrating the effectiveness of the proposed method.

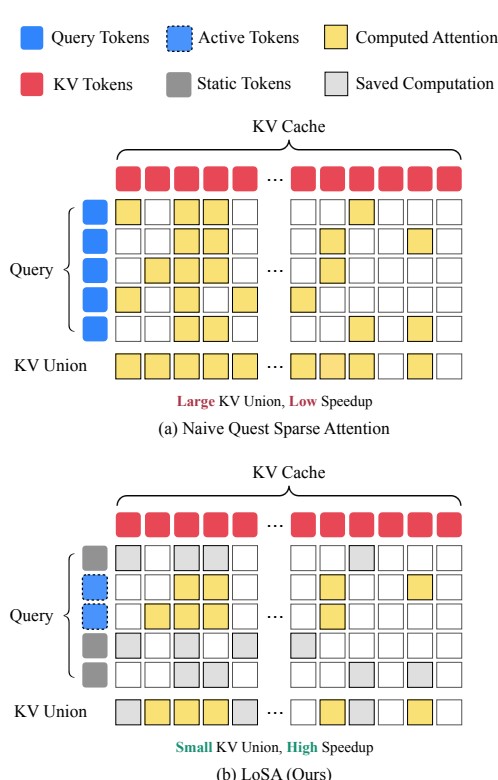

*Figure 1.* Illustration of the KV-union effect: in block diffusion, each query selects a small set of prefix KV positions, but the effective cost is determined by the union across the block, inflating KV access. Our method computes sparse attention only for selected (active) tokens during denoising and reuses cached results for the others (static tokens), substantially reducing the size of the KV union and the latency.

## 1. Introduction

Diffusion language models (DLMs) have emerged as a compelling alternative to autoregressive transformers for text generation (Austin et al., 2021; Sahoo et al., 2024; Shi et al., 2024; Lou et al., 2024; Nie et al., 2025). Unlike autore-

[1]University of California, Berkeley [2]FuriosaAI [3]ICSI [4]LBNL [5]UT Austin. Correspondence to: Amir Gholami <amirgh@berkeley.edu>.

*Proceedings of the 43rd International Conference on Machine Learning*, Seoul, South Korea. PMLR 306, 2026. Copyright 2026 by the author(s).

gressive models that generate tokens sequentially, DLMs generate tokens via iterative denoising and can update a block of tokens in any order, offering potential benefits for complex reasoning tasks. Recent block-wise DLMs (Arriola et al., 2025; Cheng et al., 2025; Ye et al., 2025a) produce tokens in diffusion-based blocks, enabling any-order, non-autoregressive generation within a block while preserving autoregressive dependencies across blocks. These models rely on a KV cache to reuse prefix representations. Despite this flexibility, they remain bottlenecked by prefix attention: at each denoising step, all tokens in the block must

Timestep $t-1$: $\|q_{t-1}\|_2$     Timestep $t$: $\|q_t\|_2$     Timestep $t$: MSE$(q_{t-1}, q_t)$

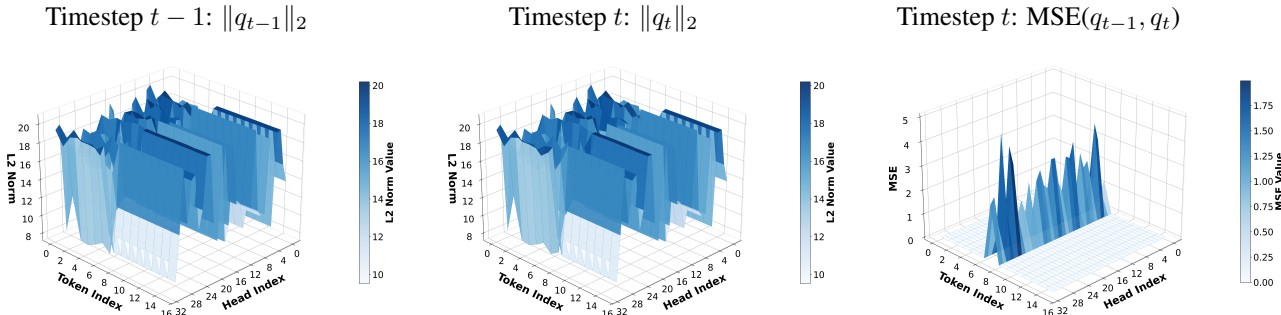

*Figure 2.* Visualizing representation change locality across denoising steps. Left and middle plots show the L2 norm of query vectors at timesteps $t-1$ and $t$ respectively. The right plot shows the per-token MSE between query vectors across the two timesteps. Only a small fraction of tokens exhibit large changes, motivating the reuse of cached-prefix attention for stable tokens.

attend to the entire cached prefix. For long contexts, the cost of repeatedly loading the KV cache dominates inference latency (Dao et al., 2022).

A natural approach to accelerate block-wise DLMs is to apply sparse attention methods developed for autoregressive LLMs (Tang et al., 2024; Zhang et al., 2023; Liu et al., 2023; Xiao et al., 2024). These methods select a subset of KV positions per query, reducing memory traffic. However, when naively adapting such methods to diffusion language models (DLMs), we encounter a **KV Inflation problem** (Figure 1), as different queries within the same block tend to select different KV positions. Since attention remains memory-bound, the overhead is determined by the size of the **union** of KV positions of all queries in the block. Let $B$ be the block size, the size of this union can increase by at most $B$ times compared with per-query budget, effectively negating the intended speedup from sparsity.

To mitigate this problem, we observe that block-wise diffusion provides a unique structure absent in autoregressive decoding: **Locality of Representation Changes** across denoising steps. As shown in Figure 2, between consecutive denoising iterations, only a small fraction of tokens undergo substantial updates to their hidden representations; we refer to these as **Active tokens**. The remaining **Stable tokens** have queries that change minimally, meaning their attention patterns, and therefore their attention outputs, remain approximately constant.

Building on this observation, we introduce LoSA (**Lo**cality-aware **S**parse **A**ttention), a method that exploits locality of representation changes to reduce KV cache memory operations. The key idea is to compute sparse prefix attention only for active tokens at each denoising step, since they are the only queries whose attention patterns change substantially. For the remaining stable tokens (i.e. tokens whose hidden representation has negligible change across denoising steps), we reuse cached prefix-attention outputs from the previous step. By reducing the number of queries that

participate in sparse attention from the block size $B$ to $|\mathcal{A}|$ (where $\mathcal{A}$ is the set of active tokens), we shrink the union of selected KV positions and directly lower memory traffic, leading to higher speedup. Beyond efficiency gains, LoSA also improves accuracy compared to naive sparse attention. The key advantage is that for stable tokens, our method preserves the full attention information from the cached prefix, whereas sparse attention can only access a subset of KV Cache. Therefore, for the majority of the tokens, LoSA attention computation is more accurate, making the performance degradation less severe.

We summarize our contributions as follows:

- We observe **Locality of Representation Changes** in block-wise diffusion: across denoising steps, only a small fraction of tokens (**Active tokens**) undergo substantial representation changes, while the majority (**Stable tokens**) maintain approximately constant. Details in § 3.2.

- We identify the **KV Inflation problem** in block-wise DLM inference: when applying sparse attention to a block of $B$ queries, the attention kernel must load the *union* of selected prefix KV positions across the block, making the speedup less than expected. Details in § 3.1.

- We propose LoSA, a sparse attention method that reuses cached prefix attention outputs for stable tokens and computes sparse attention only for active tokens, reducing the union of selected KV indices by $\sim 1.5\times$ in practice. Details in § 4.

- We evaluate LoSA on SDAR-8B (Cheng et al., 2025), Trado-8B, and Trado-4B (Wang et al., 2025a) across LongBench and reasoning benchmarks. Compared to SparseD (Wang et al., 2025b), Sparse-dLLM (Song et al., 2025), and QUEST (Tang et al., 2024), LoSA achieves up to +9 points accuracy improvement on LongBench at aggressive sparsity while maintaining $1.5\times$ lower attention density. On NVIDIA RTX A6000 GPUs, LoSA delivers up to $4.14\times$ attention speedup, with the trend transferring

to newer hardware ($3.67\times$ on RTX 5090).

## 2. Preliminaries

### 2.1. Block-Wise Diffusion Language Models

Block-wise diffusion language models (DLMs) generate text in blocks of $B$ tokens, where each block is treated as a single generation unit. Blocks are generated autoregressively, while attention within each block is bidirectional.

At a denoising step, the queries, keys and values corresponding to the current block are denoted as $\boldsymbol{Q}_b, \boldsymbol{K}_b, \boldsymbol{V}_b \in \mathbb{R}^{B \times d}$, where $d$ is the head dimension and the subscript $b$ stands for *block*. Since the denoising process follows an autoregressive manner, we maintain a KV cache of length $L$ to avoid recomputation. We denote the cached prefix keys/values as $\boldsymbol{K}_p, \boldsymbol{V}_p \in \mathbb{R}^{L \times d}$, where the subscript $p$ stands for *prefix*. Note that $\boldsymbol{K}_p$ and $\boldsymbol{V}_p$ are fixed across denoising steps, while $\boldsymbol{K}_b$ and $\boldsymbol{V}_b$ are updated at each step.

Consider a single attention head. The attention computation can be written as

$$\boldsymbol{O} = \mathrm{softmax}\left(\frac{\boldsymbol{Q}_b[\boldsymbol{K}_p, \boldsymbol{K}_b]^\top}{\sqrt{d}}\right)[\boldsymbol{V}_p, \boldsymbol{V}_b].$$

### 2.2. Attention Decomposition With Online Softmax

Starting from the full attention computation, we decompose the computation into a prefix part and a suffix part using the online-softmax method adopted in FlashAttention (Dao et al., 2022). We track each part's log-normalizer (log-sum-exp) and attention output first, then merge them together. To simplify notation, we consider a single attention head and assume the query $\boldsymbol{q} \in \mathbb{R}^d$ has only one token.

We define the score operator $\mathcal{S}$ as $\mathcal{S}(\boldsymbol{q}, \boldsymbol{K}) = \frac{\boldsymbol{q}\boldsymbol{K}^\top}{\sqrt{d}}$. Applying this operator to the prefix and current block yields score vectors $\boldsymbol{s}_p = \mathcal{S}(\boldsymbol{q}, \boldsymbol{K}_p) \in \mathbb{R}^L$ and $\boldsymbol{s}_b = \mathcal{S}(\boldsymbol{q}, \boldsymbol{K}_b) \in \mathbb{R}^B$. From these scores, we compute the corresponding log-normalizers $L$ (which is a scalar):

$$L_p = \log \sum \exp(\boldsymbol{s}_p), \qquad L_b = \log \sum \exp(\boldsymbol{s}_b).$$

The attention outputs computation can also be rewritten using the log-normalizers:

$$\boldsymbol{o}_p = \mathrm{softmax}(\boldsymbol{s}_p)\,\boldsymbol{V}_p = \frac{1}{e^{L_p}}\exp(\boldsymbol{s}_p)\,\boldsymbol{V}_p \in \mathbb{R}^d,$$

$$\boldsymbol{o}_b = \mathrm{softmax}(\boldsymbol{s}_b)\,\boldsymbol{V}_b = \frac{1}{e^{L_b}}\exp(\boldsymbol{s}_b)\,\boldsymbol{V}_b \in \mathbb{R}^d.$$

We store the local log-normalizers $L_p$ and $L_b$, together with the attention outputs $\boldsymbol{o}_p$ and $\boldsymbol{o}_b$, then merge them using the online-softmax technique to obtain the final attention output.

When the query is a matrix $\boldsymbol{Q}_b \in \mathbb{R}^{B \times d}$, the size of the log-normalizers is $\mathbb{R}^B$ and the attention outputs is $\mathbb{R}^{B \times d}$, both

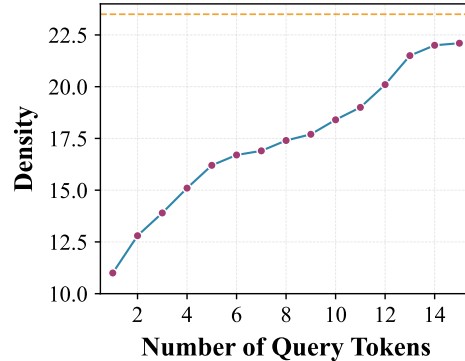

*Figure 3.* KV-cache load reduction with locality-aware sparse prefix attention. X-axis: number of active query tokens in the block; Y-axis: percentage of the full KV cache that must be loaded. Evaluated on Trado-8B-Instruct on TriviaQA with block size 16 and 64K context length. LoSA only loads prefix KV positions in the union $\mathcal{I} = \bigcup_{i \in \mathcal{A}} \mathcal{I}_i$ for the active tokens set $\mathcal{A}$, rather than the union over all $B$ queries. The dashed orange line shows the KV load under QUEST, which must load the union across all queries.

are independent of $L$, meaning that the memory complexity is minimal when context length is long.

### 2.3. Sparse Attention Algorithms for LLMs

For LLMs, sparse attention methods approximate dense attention over the prefix by selecting a small subset of prefix positions. Given query $\boldsymbol{q}_i$, a sparse selector returns an index set $\mathcal{I} \subseteq \{1, \ldots, L\}$ with budget $|\mathcal{I}| = k$, and the attention output is computed only over the selected keys/values. In this paper, we use the QUEST (Tang et al., 2024) algorithm and naively adapted it for DLMs as a baseline to compare against. QUEST partitions the KV cache into contiguous pages of size $g$ and maintains lightweight min and max keys for each page. At inference time, it computes a query-dependent upper bound on the attention score for each page and selects the top $\frac{k}{g}$ pages for exact attention computation. Since the min and max keys are $g\times$ smaller than the original KV cache, the selection overhead is $\frac{g}{2}\times$ smaller than dense attention, which is minimal when the page size is large. Note that LoSA (introduced in § 4) is largely orthogonal to the choice of sparse selector algorithm.

## 3. Motivation

### 3.1. KV Inflation Problem

In autoregressive LLM decoding, sparse attention methods restrict each query to attend to only $k$ KV positions, making the attention latency roughly proportional to $k$. As a result, reducing the number of KV vectors loaded generally yields proportional speedup.

However, for DLMs, the situation is different. Suppose at each denoising step, the model processes a *block* of $B$ query

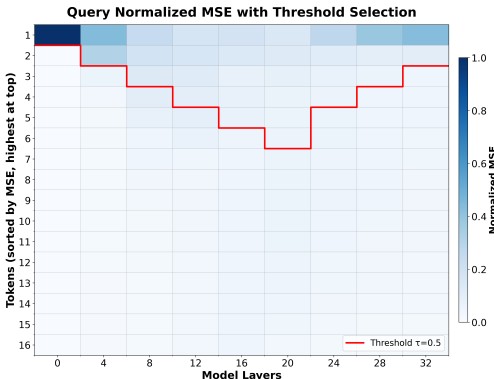

*Figure 4.* Visualizing locality across denoising steps. We plot the distribution of per-token MSE changes in queries between steps $t-1$ and $t$. For each layer, we sort the tokens in decreasing order of MSE changes from top to bottom in this heatmap. We observe that only a small fraction of tokens exhibit large changes (we show a threshold line which corresponds to 50% of the total MSE change is exhibited by tokens above that line), motivating the reuse of cached-prefix attention for the stable tokens.

tokens. The attention workload consists of $B$ query tokens attending to $(L + B)$ KV cache tokens. A naive adaptation of sparse attention applies a fixed budget $k$ *per query token* in the block, producing index sets $\{\mathcal{I}_i\}_{i=1}^{B}$ with $|\mathcal{I}_i| = k$ for each query $i$. To ensure all important information is captured, we compute the union of selected KV positions across all queries. The effective number of KV positions being loaded is denoted as $L'$:

$$L' = \left| \bigcup_{i=1}^{B} \mathcal{I}_i \right|.$$

Since the block size $B$ is typically not large enough to make attention compute-bound, $L'$ directly determines the memory traffic and thus the attention computation latency. The key insight is that loading a KV position incurs similar overhead whether it is used by one query or all queries in the block, making $L'$ the critical factor for performance.

In this case, we identify a **KV Inflation problem** which makes sparse attention extremely inefficient for DLM workloads. Even though each query selects only selected $k$ KV positions, different queries may select different positions. Therefore, $L'$ can become much larger than the intended budget $k$, negating the intended speedup from sparsity. This phenomenon is well illustrated in Figure 1.

In the worst case, $L'$ can grow to $\min(L, Bk)$. This large union size increases memory traffic and leads to sub-optimal speedup performance. As indicated in Figure 3, the KV Inflation problem can lead to $\sim 2.0\times$ higher attention density.

## 3.2. Locality of Representation Changes Across Denoising Steps

Our solution begins by observing that block diffusion decoding offers an additional structure that autoregressive LLM decoding does not have: *locality of representation changes across denoising steps*. Between two consecutive denoising steps $t-1$ and $t$, only a small subset of tokens in the current block are updated from [MASK] to actual tokens (see Figure 2). Tokens surrounding these updated positions experience significant changes in their hidden representations, while the remaining tokens' hidden states remain approximately constant.

Based on this observation, we categorize tokens into two groups: **active tokens**, which undergo substantial changes in their hidden representations, and **stable tokens**, which have approximately constant hidden states across consecutive denoising steps. We refer to this phenomenon as **Locality of Representation Changes**.

**Quantifying locality for each token.** To measure locality, we track the per-token change in queries, keys, and values using mean squared error (MSE). For $\boldsymbol{X} \in \{\boldsymbol{Q}, \boldsymbol{K}_b, \boldsymbol{V}_b\}$, let $\boldsymbol{x}^{(t)} \in \mathbb{R}^{B \times d}$ denote hidden states at denoising step $t$. We define the locality score as the per-token MSE between the current and previous step.

$$\Delta_x^{(t)} = \mathrm{mean}\left( \frac{1}{d} \left\| \boldsymbol{x}^{(t)} - \boldsymbol{x}^{(t-1)} \right\|_2^2 \right)_{\text{token axis}} \in \mathbb{R}^B. \quad (1)$$

We visualize the distribution of $\Delta_x^{(t)}$ across tokens in Figure 2. The visualization confirms that active tokens exhibit high locality scores, while stable tokens have small scores. Notably, tokens at positions being decoded in the previous step show the largest changes.

We also observe that locality is layer-dependent. As shown in Figure 4, we sort the tokens by their locality scores and plot the distribution of locality scores for each layer. We observe that early layers and later layers tend to be more localized than middle layers.

## 4. Methodology

In this section, we describe the methodology of our LoSA method based on the aforementioned observations.

### 4.1. Locality Pruning

Let $\Delta \in \mathbb{R}^B$ denote the per-token locality score vector defined in Equation 1. We rank tokens by their locality scores in descending order and select the Top-$k$ tokens with the largest changes, forming the set $\mathcal{A}$ of **active tokens**. Tokens not in $\mathcal{A}$ are regarded as **stable tokens**.

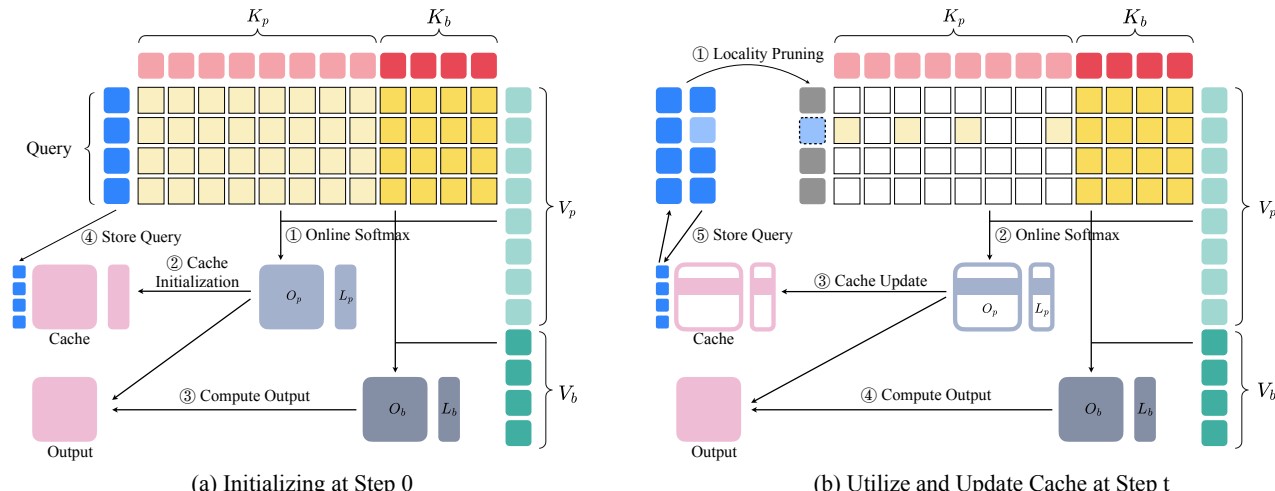

(a) Initializing at Step 0          (b) Utilize and Update Cache at Step t

*Figure 5.* Overview of the locality-aware sparse prefix attention workflow. We (i) measure per-token query changes and select the changed-token set $\mathcal{A}$, (ii) run QUEST to obtain per-token prefix indices and take their union $\mathcal{I}$, (iii) load prefix KV for $\mathcal{I}$ and compute updated prefix statistics for queries in $\mathcal{A}$ while reusing cached statistics for stable tokens, (iv) compute dense within-block attention, and (v) merge prefix and block contributions via online softmax and cache the updated prefix statistics for the next denoising step.

## 4.2. Locality-Aware Sparse Attention for Prefix Tokens

We then propose a locality-aware sparse attention algorithm to reduce the KV cache loads. We cache and reuse the attention outputs for stable tokens, and only compute the attention for active tokens using sparse attention.

**Reuse attention outputs for stable tokens.** For stable tokens $i \notin \mathcal{A}$, since the query hidden states remain stable, the attention outputs for these tokens are also stable. Therefore, we can reuse the prefix-attention statistics $(\boldsymbol{o}_p, L_p)$ from the previous step:

$$(\boldsymbol{o}_p, L_p) \leftarrow (\boldsymbol{o}_p^{\mathrm{prev}}, L_p^{\mathrm{prev}}), \quad \text{token} \notin \mathcal{A}.$$

**Sparse attention with smaller union size for active tokens.** For active tokens $i \in \mathcal{A}$, we apply the same sparse selector (QUEST) with budget $k$ to select the prefix indices, but *only for the active tokens in $\mathcal{A}$*:

$$\mathcal{I}_i = \mathrm{QUEST}(\boldsymbol{q}_i, \boldsymbol{K}_p; k), \quad i \in \mathcal{A}, \quad \mathcal{I} = \bigcup_{i \in \mathcal{A}} \mathcal{I}_i.$$

We load the prefix key/value vectors indexed by the union set $\mathcal{I}$. Then, for all queries in $\mathcal{A}$, we compute prefix attention over this shared set $\mathcal{I}$. After the computation, we update the prefix statistics $(\boldsymbol{o}_p, L_p)$ for these tokens:

$$(\boldsymbol{o}_p, L_p) \leftarrow (\boldsymbol{o}_p^{\mathrm{current}}, L_p^{\mathrm{current}}), \quad \text{token} \in \mathcal{A}.$$

**Initialization Workflow.** For the first denoising iteration for a block, cached statistics are unavailable; we fall back to dense prefix attention for initialization and start to apply

the locality-aware sparse attention from the second iteration onward. We find that this initialization step is crucial for the performance of LoSA. The reason is that the dense prefix attention is more accurate than the sparse prefix attention, and the stable tokens can benefit from the cached attention outputs by capturing all information in the prefix.

## 4.3. Computation of the Suffix Tokens and Final Output

We next focus on the computation of $\boldsymbol{K}_b$ and $\boldsymbol{V}_b$. Although keys and values also exhibit the same locality structure as queries, their relevant attention computation is minimal since $B \ll L$. Therefore, we compute the attention between $Q_b$, $K_b$, and $V_b$ densely, and get the low-dimensional statistics $(\boldsymbol{o}_b, L_b)$ for suffix tokens.

Finally, we merge the prefix and current block contributions using the online-softmax merge to obtain the final attention output. We cache the updated prefix statistics $(\boldsymbol{o}_p, L_p)$, together with the query states for the next denoising step.

## 4.4. Why Locality Reduces KV Loads: Union Size

The KV-cache traffic for sparse prefix attention is governed by the number of unique prefix positions (or KV tiles) that must be fetched: $|\mathcal{I}| = \left|\bigcup_{i \in \mathcal{A}} \mathcal{I}_i\right|$. Compared to the naive per-token sparse scheme (which uses $\mathcal{A} = \{1, \ldots, B\}$), our locality-aware scheme uses a smaller set of queries, which can only reduce (never increase) the union:

$$\left|\bigcup_{i \in \mathcal{A}} \mathcal{I}_i\right| \leq \left|\bigcup_{i=1}^{B} \mathcal{I}_i\right| \leq \min(L, |\mathcal{A}|k).$$

| KV Cache Per-Query Budget | Method | LongBench Accuracy (Trado-8B-Instruct) | | | | | |
|---|---|---|---|---|---|---|---|
| | | HotPotQA | TriviaQA | NarrativeQA | Qasper | MultiFieldQA | Average $_{(\uparrow)}$ |
| - | Dense | 49.45% | 84.79% | 19.04% | 17.75% | 53.29% | 44.86% |
| 128 | QUEST | 29.17% | 69.21% | 7.46% | 13.64% | 38.21% | 31.54% |
| | LoSA | 48.27% | 83.79% | 17.19% | 15.58% | 45.00% | **41.97%**$_{(+10.43)}$ |
| 256 | QUEST | 32.95% | 75.23% | 8.50% | 14.04% | 42.46% | 34.64% |
| | LoSA | 44.53% | 81.82% | 19.42% | 17.11% | 47.81% | **42.14%**$_{(+7.50)}$ |
| 512 | QUEST | 34.58% | 79.33% | 8.99% | 16.57% | 44.04% | 36.70% |
| | LoSA | 44.19% | 84.97% | 18.39% | 13.30% | 50.14% | **42.20%**$_{(+5.50)}$ |
| 1024 | QUEST | 39.88% | 78.84% | 7.37% | 17.35% | 45.00% | 37.69% |
| | LoSA | 48.45% | 82.32% | 18.89% | 15.78% | 48.94% | **42.88%**$_{(+5.19)}$ |

*Table 1.* Accuracy (%) on LongBench under different retrieval budgets for **Trado-8B-Instruct**. LoSA consistently outperforms QUEST across all budget settings, with particularly large gains at low budgets (e.g., +10.43 at budget 128). The $\Delta$ shown is the improvement over QUEST. Full comparison including SparseD is in Table 6.

| KV Cache Per-Query Budget | Method | LongBench KV-Cache Density | | | | | |
|---|---|---|---|---|---|---|---|
| | | HotPotQA | TriviaQA | NarrativeQA | Qasper | MultiFieldQA | Average $_{(\downarrow)}$ |
| 128 | QUEST | 2.93% | 3.80% | 1.90% | 6.72% | 5.87% | 4.24% |
| | LoSA | 1.71% | 2.29% | 0.98% | 4.15% | 3.77% | 2.58%$_{(1.64\times)}$ |
| 256 | QUEST | 5.84% | 7.07% | 3.89% | 12.47% | 11.14% | 8.08% |
| | LoSA | 3.37% | 4.47% | 2.05% | 7.90% | 7.43% | 5.04%$_{(1.60\times)}$ |
| 512 | QUEST | 10.74% | 13.06% | 7.24% | 23.13% | 19.71% | 14.78% |
| | LoSA | 6.63% | 8.54% | 3.99% | 15.39% | 14.00% | 9.71%$_{(1.52\times)}$ |
| 1024 | QUEST | 19.10% | 22.85% | 13.06% | 39.06% | 34.21% | 25.66% |
| | LoSA | 12.57% | 16.24% | 7.50% | 28.81% | 26.04% | 18.23%$_{(1.41\times)}$ |

*Table 2.* KV-cache density (percentage of prefix tokens selected) on LongBench under different budgets. On all datasets and budget configurations, LoSA achieves an average of $1.54\times$ lower KV-cache density than the baseline.

The reduction is generally *not* proportional to $|\mathcal{A}|$: it depends on the overlap structure among the $\{\mathcal{I}_i\}$. Nevertheless, shrinking the participating query set from $B$ to $|\mathcal{A}|$ eliminates many query-induced selections and typically decreases the number of KV cache that must be loaded. As shown in Figure 3, a smaller union size leads to smaller KV cache loads and higher speedup.

### 4.5. Efficiency Analysis

The overhead introduced by LoSA consists of three main components: (1) locality pruning to identify active tokens, (2) sparse selection to select the prefix indices that participate in the sparse attention computation and compute its union, and (3) computing the final attention output.

The locality pruning step requires calculating the per-token locality scores and ranking tokens by their scores, which incurs $\mathcal{O}(B \times d + B \log B)$ FLOPs overhead. The sparse selector incurs $\mathcal{O}(\frac{B}{g} \times d)$ overhead, where $g$ is the page size used by QUEST. Computing their union incurs $\mathcal{O}(\frac{B}{g})$ overhead. The final attention output computation also incurs $\mathcal{O}(B \times d)$ overhead, as the online-softmax merge is applied

to LSE and local attention outputs. The total overhead is $\mathcal{O}(B \times d + \frac{B}{g} \times d)$. Since $B \ll L$ in typical DLM settings, these overheads are minimal compared to the actual attention computation.

**Memory overhead.** The additional memory required by LoSA is minimal. We cache the prefix attention output $\boldsymbol{o}_p \in \mathbb{R}^{B \times d}$ and the log-normalizer $L_p \in \mathbb{R}^B$ for each attention head, yielding $B(d+1)$ extra scalars per head. For typical settings ($B = 16$, $d = 128$), this amounts to $\sim 2$ KB per head in FP16, which is negligible compared to the KV cache size $\mathcal{O}(L \times d)$. The remaining memory traffic is dominated by loading the union of selected KV positions, which LoSA directly reduces.

## 5. Experiments and Results

### 5.1. Experimental Settings

**Models.** We evaluate LoSA on three block-wise diffusion language models: Trado-8B-Instruct, Trado-4B-Instruct (Wang et al., 2025a), and SDAR-8B-Instruct (Cheng et al., 2025). All models follow the semi-autoregressive

| KV Cache | | LongBench Accuracy (SDAR-8B-Chat) | | | | | |
|---|---|---|---|---|---|---|---|
| Per-Query Budget | Method | HotPotQA | TriviaQA | NarrativeQA | Qasper | MultiFieldQA | Average $_{(\uparrow)}$ |
| - | Dense | 49.35% | 85.72% | 19.06% | 18.25% | 49.49% | 44.37% |
| 128 | QUEST | 27.31% | 70.40% | 6.44% | 17.29% | 41.40% | 32.57% |
| | LoSA | 43.36% | 80.32% | 18.69% | 15.63% | 46.74% | **40.95%** $_{(+8.38)}$ |
| 256 | QUEST | 32.57% | 78.36% | 8.73% | 18.21% | 43.20% | 36.21% |
| | LoSA | 45.68% | 83.16% | 15.65% | 13.17% | 48.33% | **41.20%** $_{(+4.99)}$ |
| 512 | QUEST | 32.50% | 78.43% | 9.80% | 19.36% | 43.08% | 36.63% |
| | LoSA | 47.77% | 83.29% | 17.37% | 14.12% | 49.90% | **42.49%** $_{(+5.86)}$ |
| 1024 | QUEST | 32.59% | 80.92% | 11.00% | 19.28% | 42.30% | 37.22% |
| | LoSA | 47.84% | 85.93% | 18.44% | 14.70% | 48.43% | **43.07%** $_{(+5.85)}$ |

*Table 3.* Accuracy (%) on LongBench under different retrieval budgets for **SDAR-8B-Chat**. Results are consistent with Trado-8B (Table 1): LoSA consistently outperforms QUEST, with the largest gains at low budgets (+8.38 at budget 128). Full comparison including SparseD is in Table 6.

| | | Block Size = 16 | | | | | | Block Size = 32 | | | | | |
|---|---|---|---|---|---|---|---|---|---|---|---|---|---|
| Budget | Method | HotPotQA | TriviaQA | NarrativeQA | Qasper | MFieldQA | Avg. $_{(\uparrow)}$ | HotPotQA | TriviaQA | NarrativeQA | Qasper | MFieldQA | Avg. $_{(\uparrow)}$ |
| - | Dense | 38.16% | 75.61% | 18.80% | 29.57% | 35.42% | 39.51% | 36.22% | 75.23% | 19.58% | 13.40% | 31.52% | 35.19% |
| 128 | QUEST | 30.01% | 69.14% | 9.83% | 26.05% | 30.79% | 33.16% | 25.55% | 70.83% | 9.53% | 11.20% | 27.30% | 28.88% |
| | LoSA | 33.65% | 66.46% | 14.78% | 24.71% | 30.07% | **33.93%** | 28.62% | 69.50% | 12.15% | 13.78% | 28.25% | **30.46%** |
| 256 | QUEST | 33.06% | 73.49% | 12.34% | 29.25% | 34.67% | 36.56% | 31.61% | 72.10% | 10.87% | 12.85% | 29.10% | 31.31% |
| | LoSA | 36.33% | 72.74% | 15.99% | 28.11% | 34.27% | **37.49%** | 32.14% | 72.20% | 15.99% | 14.80% | 28.68% | **32.76%** |

*Table 4.* LongBench accuracy (%) for **Trado-4B-Instruct** under two block sizes. LoSA consistently outperforms QUEST across model sizes and block-size configurations, confirming that the locality-aware approach generalizes beyond the 8B scale.

| Budget | Method | HellaSwag | WinoGrande | BoolQ | Avg. |
|---|---|---|---|---|---|
| – | Dense | 69.60 | 63.60 | 72.92 | 68.71 |
| 128 | QUEST | 67.60 | 66.80 | 73.96 | 69.45 |
| | LoSA | 70.80 | 66.00 | 72.92 | 69.91 |
| 256 | QUEST | 73.60 | 63.20 | 75.00 | 70.60 |
| | LoSA | 70.40 | 65.20 | 73.96 | 69.85 |

*Table 5.* Accuracy (%) on three commonsense reasoning benchmarks for Trado-8B-Instruct. LoSA performs comparably to QUEST; the slight gap at budget 256 is within statistical variance on these short-context tasks (<1K tokens) where the KV Inflation problem is less severe.

block-wise generation paradigm. We set the block size to 16 if not specified.

**Baselines.** We compare against QUEST (Tang et al., 2024), a per-query sparse attention method originally designed for autoregressive LLMs, which we adapt for DLMs by applying it independently to each query in a block. We additionally compare with SparseD (Wang et al., 2025b) in Table 6.

**Datasets.** We evaluate mainly on LongBench (Bai et al., 2024) to examine their long-context ability. To be specific, we evaluate on HotPotQA (Yang et al., 2018), TriviaQA (Joshi et al., 2017), NarrativeQA (Kočiský et al., 2018), Qasper (Dasigi et al., 2021), MultiFieldQA (Bai et al., 2024), and their average. We also evaluate on commonsense reason-

ing benchmarks, including HellaSwag (Zellers et al., 2019), WinoGrande (Sakaguchi et al., 2021), and BoolQ (Clark et al., 2019).

**Implementation** We implement our algorithm using customized CUDA and Triton (Tillet & Cox, 2019) kernels, together with attention kernels from FlashInfer (Ye et al., 2025b). We pick the top-5 query tokens in our locality pruning algorithm.

### 5.2. LongBench Results

We evaluate LoSA on LongBench across multiple models. As shown in Table 1, on Trado-8B-Instruct, LoSA consistently outperforms QUEST across all budget configurations and datasets. At budget 128, LoSA achieves 41.97% average accuracy, outperforming QUEST (31.54%) by +10.43. At budget 256, LoSA (42.14%) leads QUEST (34.64%) by +7.50. At higher budgets (512 and 1024), both methods converge toward dense accuracy, with LoSA maintaining a clear advantage. Notably, LoSA maintains accuracy close to the Dense baseline even at aggressive sparsity levels, demonstrating that locality-aware reuse effectively preserves important attention information. In terms of attention density, LoSA maintains an average of $1.54\times$ lower attention density than QUEST across all budget configurations and datasets (Table 2).

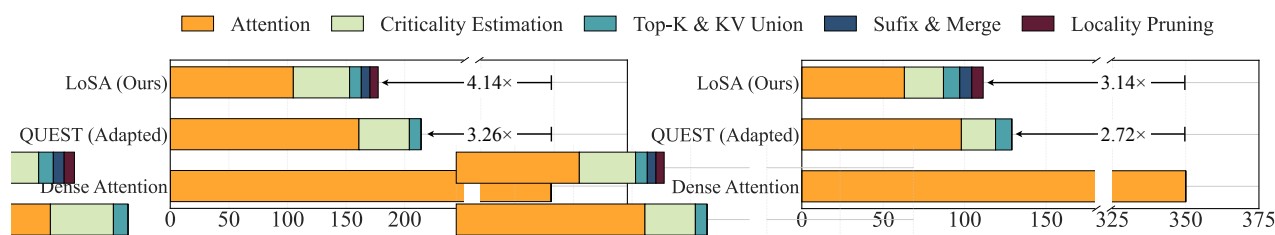

*Figure 6.* Latency breakdown comparison on prefix attention using Trado-8B-Instruct on TriviaQA with RTX A6000. **Left:** 64K context length with block size 16. **Right:** 32K context length with block size 32. LoSA and QUEST (Adapted) achieve 4.14× and 3.26× speedup over Dense Attention, respectively, by reducing memory-bound attention operations through sparsity. LoSA further improves over QUEST through locality-aware reuse.

On SDAR-8B-Chat (Table 3), the trends are consistent: LoSA outperforms QUEST at all budget levels, with the largest gains at budget 128 (+8.38). At higher budgets (512, 1024), both methods converge toward dense accuracy.

To verify that LoSA generalizes across model sizes, we further evaluate on Trado-4B-Instruct with both block size 16 and block size 32 (Table 4). LoSA consistently outperforms QUEST across both configurations, confirming that the locality-aware approach is not specific to one model size or block-size setting.

### 5.3. Commonsense Reasoning Results

We evaluate LoSA on commonsense reasoning benchmarks using Trado-8B-Instruct. Table 5 shows accuracy results under different budgets. LoSA achieves competitive or better accuracy compared to QUEST across all three datasets. At budget 128, LoSA achieves an average accuracy of 69.91%, slightly outperforming QUEST's 69.45%. At budget 256, QUEST achieves 70.60% average accuracy while LoSA maintains 69.85%; this slight gap is within statistical variance and is expected since these short-context benchmarks (typically <1K tokens) do not strongly exhibit the KV Inflation problem that LoSA is designed to address. LoSA's advantage is most pronounced in long-context settings (Table 1).

### 5.4. Latency Analysis

We measure end-to-end latency of prefix attention computation to validate that our method translates sparsity into wall-clock speedup. We evaluate two configurations: (1) 64K context length with 16 token block size, and (2) 32K context length with 32 token block size, both using Trado-8B-Instruct and testing on TriviaQA.

Figure 6 compares the latency breakdown of LoSA, QUEST, and Dense Attention. The latency breakdown consists of several components: (1) **Criticality Estimation**, which refers to the sparse attention selector (QUEST) that identifies relevant KV positions for each query; (2) **Locality Pruning**, which identifies active tokens by computing

locality scores based on representation changes; (3) **Cache Management**, which stores and retrieves cached attention outputs for stable tokens; and (4) **Attention Computation**, which performs the actual sparse attention operations over selected KV positions.

On the RTX A6000, LoSA achieves a 4.14× speedup over Dense Attention, outperforming QUEST's 3.26× speedup. The key advantage of LoSA is that locality-aware reuse substantially reduces the number of queries requiring fresh attention computation. By caching and reusing attention outputs for stable tokens, LoSA minimizes memory-bound operations while maintaining accuracy, leading to lower attention computation time despite the additional overhead of locality pruning and cache management.

**Scalability to newer GPUs.** We additionally measure latency on an RTX 5090 under the 64K context length, block size 16 setting. Dense attention takes 650 $\mu$s, while LoSA completes in 177 $\mu$s (3.67× speedup) and QUEST in 228 $\mu$s (2.85× speedup). The speedup trend transfers across GPU generations because the workflow remains memory-bound: the dominant cost is KV data movement rather than compute. As memory bandwidth improves on newer hardware, kernel launch overhead and other fixed costs become a larger relative fraction, leading to slightly lower but still substantial speedups.

## 6. Related Work

### 6.1. Diffusion Language Models

Diffusion models have achieved remarkable success in continuous domains such as images and audio, inspiring efforts to adapt them for discrete text generation, which enables any-order decoding (Kang et al., 2025). Early work explored discrete diffusion through absorbing states (Austin et al., 2021) and multinomial diffusion processes. More recently, masked diffusion language models (MDLM) (Sahoo et al., 2024) demonstrated that simple masked diffusion with modern architectures can match autoregressive baselines on language modeling benchmarks. SEDD (Lou et al.,

2024) introduced score entropy-based training for discrete diffusion, achieving state-of-the-art perplexity. LLaDA (Nie et al., 2025) and Dream (Ye et al., 2025a) scaled masked diffusion models demonstrating competitive performance with autoregressive models of similar scale.

**Block-wise diffusion models.** Recent work has explored semi-autoregressive diffusion that generates tokens in blocks (Kang et al., 2025). BD3-LMs (Arriola et al., 2025) interpolate between fully autoregressive and fully any-order generation by producing fixed-size blocks autoregressively while using diffusion within each block. SDAR (Cheng et al., 2025) converts an autoregressive LLM into a block-diffusion model. Trado (Wang et al., 2025a) applies advanced post-training techniques to enhance their reasoning capabilities. Efforts to extend DLMs to longer contexts include LongLLaDA (Liu et al., 2025), which unlocks long-context capabilities in diffusion LLMs, and UltraL-LaDA (He et al., 2025), which scales context length to 128K tokens. Our work is designed for this block-wise setting.

**Efficient inference for DLMs.** Several concurrent works address inference acceleration in DLMs. Fast-dLLM (Wu et al., 2025b) and its successor Fast-dLLM v2 (Wu et al., 2025a) enable KV caching and any-order decoding for diffusion LLMs without additional training. SparseD (Wang et al., 2025b) observes that sparse attention patterns change little across denoising steps and reuses a fixed sparse pattern computed at initialization. Sparse-dLLM (Song et al., 2025) accelerates diffusion LLMs by dynamically evicting KV cache entries. Our method differs fundamentally from these approaches: LoSA reuses *attention outputs* (not patterns) for stable tokens, allowing them to retain full dense-attention information, and applies dynamic sparse attention only to active tokens whose representations have changed.

### 6.2. Sparse Attention for Long-Context LLMs

Reducing the quadratic complexity of attention has been extensively studied. Fixed sparse patterns such as local windows combined with global tokens (Child et al., 2019; Beltagy et al., 2020; Zaheer et al., 2020) reduce complexity to linear but require architectural modifications during training. Content-based sparse attention methods like Reformer (Kitaev et al., 2020) use locality-sensitive hashing to identify relevant keys. Linear attention variants (Katharopoulos et al., 2020; Choromanski et al., 2021) approximate softmax attention with kernel feature maps, enabling recurrent computation.

For pretrained LLMs at inference time, dynamic sparse attention methods select relevant KV positions per query without retraining. QUEST (Tang et al., 2024) groups KV cache into pages and selects pages based on estimated attention scores. $H_2O$ (Zhang et al., 2023) maintains a dynamic cache of "heavy hitter" tokens that accumulate high attention mass. Scissorhands (Liu et al., 2023) exploits the persistence of important tokens across generation steps. StreamingLLM (Xiao et al., 2024) discovers that keeping initial "sink" tokens enables stable streaming inference. SnapKV (Li et al., 2024) and PyramidKV (Cai et al., 2024) compress the KV cache based on attention patterns observed during prefill. MInference (Jiang et al., 2024) accelerates long-context prefilling through dynamic sparse patterns.

### 6.3. KV Cache Optimization and Efficient Serving

Beyond sparse attention, several orthogonal techniques improve KV cache efficiency. FlashAttention (Dao et al., 2022; Dao, 2024) optimizes memory access patterns through tiling, reducing memory traffic for dense attention without approximation. PagedAttention (Kwon et al., 2023) enables non-contiguous KV cache storage, improving memory utilization in serving systems. SageAttention (Zhang et al., 2025c;a;b;d) quantizes the attention computation to lower precision to speedup attention further.

## 7. Limitations

While LoSA is effective for long-context block-wise DLM inference, its advantage diminishes for short-context inputs (e.g., <1K tokens) where the KV Inflation problem is less severe and all methods perform comparably. Additionally, our current evaluation focuses on batch size 1; integrating LoSA with batched serving systems (e.g., vLLM, TensorRT-LLM) is an interesting direction for future work. Finally, LoSA requires an initial dense attention pass at the first denoising step of each block, which adds a fixed overhead that becomes amortized over subsequent denoising iterations.

## 8. Conclusion

In this paper, we identify the KV Inflation problem in block-wise diffusion language models as the main reason why naive sparse attention fails to accelerate block-wise DLMs. We observe locality of representation changes across denoising steps and propose LoSA, which reuses cached attention for stable tokens and computes sparse attention only for active tokens. This reduces the number of queries participating in sparse KV selection, effectively mitigating the KV Inflation problem. Across Trado-8B, SDAR-8B, and Trado-4B, LoSA consistently outperforms SparseD, Sparse-dLLM, and QUEST on LongBench, with up to +9.01 average accuracy improvement at aggressive sparsity (budget 128). LoSA also achieves $1.54\times$ lower attention density and up to $4.14\times$ speedup on A6000 GPUs, with the speedup trend transferring to newer hardware ($3.67\times$ on RTX 5090).

## Impact Statement

This paper presents work that aims to advance the field of Machine Learning. There are many potential societal consequences of our work, none of which we feel must be specifically highlighted here.

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

# A. Appendix

## A.1. Full LongBench Comparison with SparseD

| KV Cache Per-Query Budget | Method | LongBench Accuracy – Comparison with SparseD | | | | | |
|---|---|---|---|---|---|---|---|
| | | HotPotQA | TriviaQA | NarrativeQA | Qasper | MultiFieldQA | Average (↑) |
| **Trado-8B-Instruct** | | | | | | | |
| - | Dense | 49.45% | 84.79% | 19.04% | 17.75% | 53.29% | 44.86% |
| 128 | QUEST | 29.17% | 69.21% | 7.46% | 13.64% | 38.21% | 31.54% |
| | SparseD | 38.97% | 64.64% | 13.70% | 7.95% | 39.55% | 32.96% |
| | LoSA | 48.27% | 83.79% | 17.19% | 15.58% | 45.00% | **41.97%**(+9.01) |
| 256 | QUEST | 32.95% | 75.23% | 8.50% | 14.04% | 42.46% | 34.64% |
| | SparseD | 43.84% | 73.63% | 15.33% | 13.26% | 49.44% | 39.10% |
| | LoSA | 44.53% | 81.82% | 19.42% | 17.11% | 47.81% | **42.14%**(+3.04) |
| 512 | QUEST | 34.58% | 79.33% | 8.99% | 16.57% | 44.04% | 36.70% |
| | SparseD | 47.84% | 80.78% | 14.08% | 13.85% | 52.16% | 41.74% |
| | LoSA | 44.19% | 84.97% | 18.39% | 13.30% | 50.14% | **42.20%**(+0.46) |
| 1024 | QUEST | 39.88% | 78.84% | 7.37% | 17.35% | 45.00% | 37.69% |
| | SparseD | 48.98% | 80.25% | 19.43% | 15.77% | 50.11% | 42.91% |
| | LoSA | 48.45% | 82.32% | 18.89% | 15.78% | 48.94% | 42.88% |
| **SDAR-8B-Chat** | | | | | | | |
| - | Dense | 49.35% | 85.72% | 19.06% | 18.25% | 49.49% | 44.37% |
| 128 | QUEST | 27.31% | 70.40% | 6.44% | 17.29% | 41.40% | 32.57% |
| | SparseD | 42.09% | 63.08% | 12.94% | 9.04% | 36.24% | 32.68% |
| | LoSA | 43.36% | 80.32% | 18.69% | 15.63% | 46.74% | **40.95%**(+8.27) |
| 256 | QUEST | 32.57% | 78.36% | 8.73% | 18.21% | 43.20% | 36.21% |
| | SparseD | 48.39% | 73.28% | 15.83% | 14.84% | 47.37% | 39.94% |
| | LoSA | 45.68% | 83.16% | 15.65% | 13.17% | 48.33% | **41.20%**(+1.26) |
| 512 | QUEST | 32.50% | 78.43% | 9.80% | 19.36% | 43.08% | 36.63% |
| | SparseD | 50.52% | 81.83% | 15.44% | 18.52% | 48.93% | 43.05% |
| | LoSA | 47.77% | 83.29% | 17.37% | 14.12% | 49.90% | 42.49% |
| 1024 | QUEST | 32.59% | 80.92% | 11.00% | 19.28% | 42.30% | 37.22% |
| | SparseD | 48.25% | 83.44% | 15.60% | 16.15% | 48.97% | 42.48% |
| | LoSA | 47.84% | 85.93% | 18.44% | 14.70% | 48.43% | **43.07%**(+0.59) |

*Table 6.* Full LongBench accuracy comparison including SparseD. The Δ shown is the improvement of LoSA over the best competing method at each budget. At budget 1024 on Trado-8B and budget 512 on SDAR-8B, SparseD slightly outperforms LoSA in average accuracy.

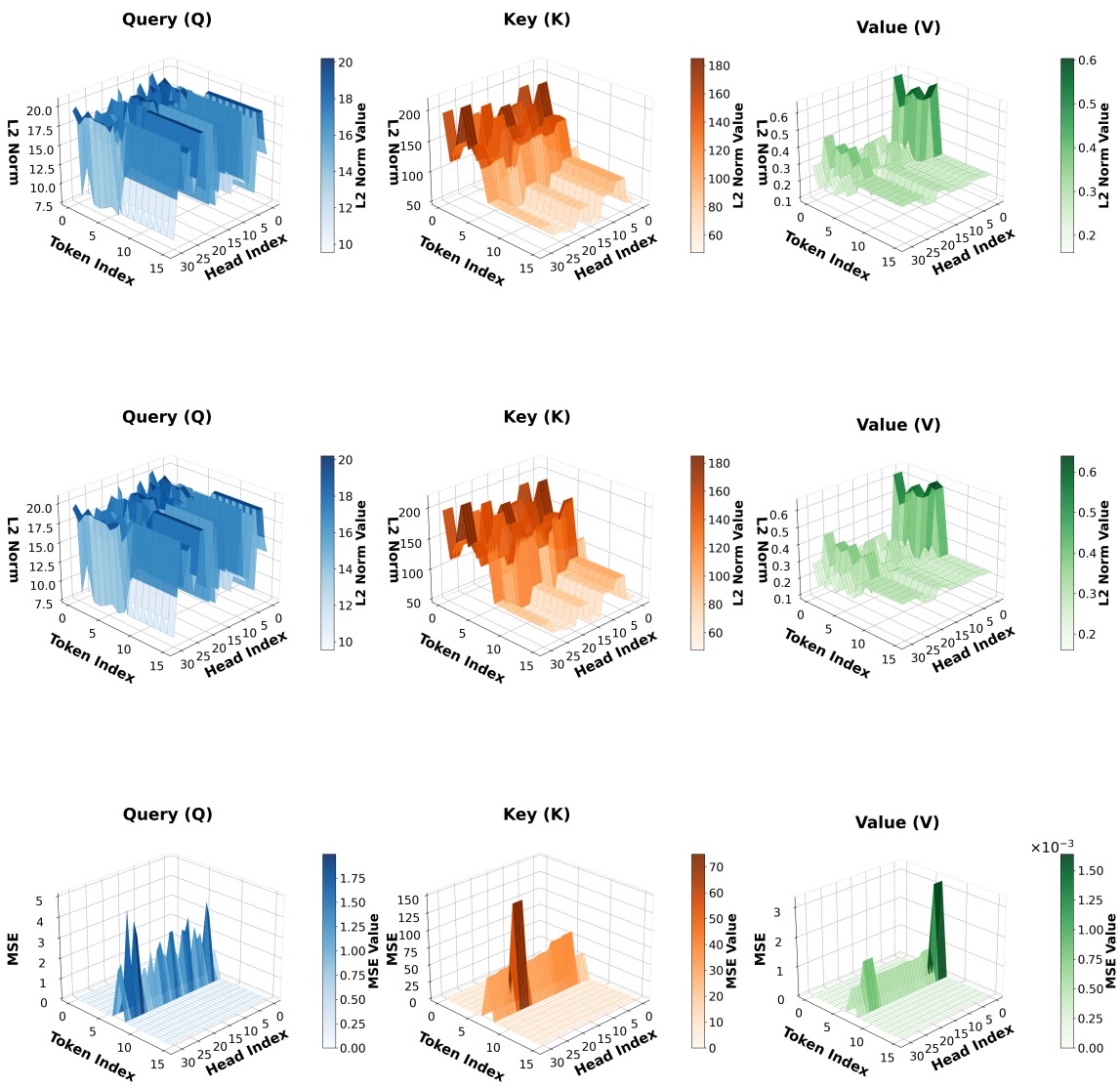

*Figure 7.* Visualizing representation change locality across denoising steps for query, keys and values of the token in the block that is being decoded. **(Top)** L2 norm of vectors at step $t-1$. **(Middle)** L2 norm of vectors at step $t$. **(Bottom)** Per-token change in representations, measured using MSE between steps $t-1$ and $t$. Only a small fraction of tokens exhibit large changes, motivating reuse of cached-prefix attention for stable tokens.

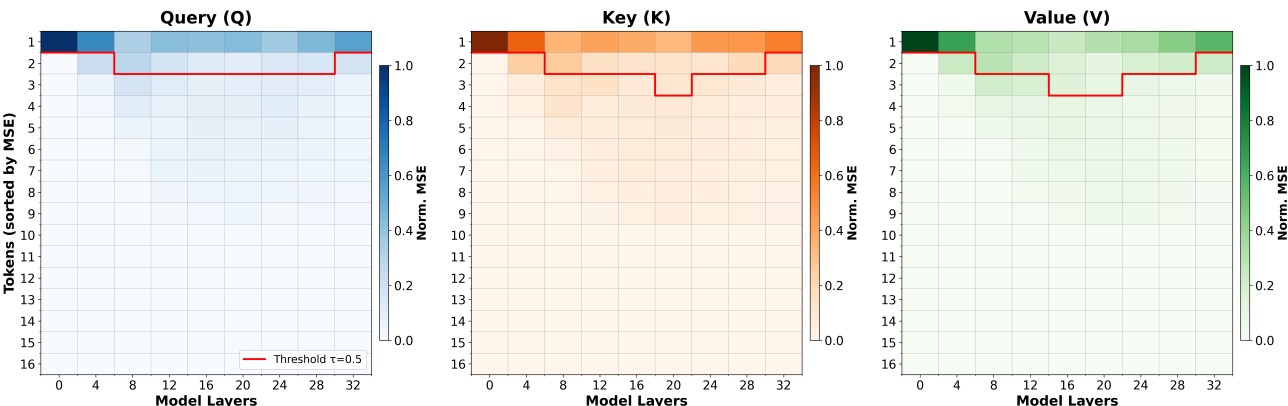

*Figure 8.* Combined QKV Normalized MSE Heatmap with Threshold Selection ($\tau = 0.5$) for Data Sample 0. The red line indicates the boundary where cumulative normalized MSE reaches 50% of the layer total. Tokens above the line are selected; tokens below are dimmed.

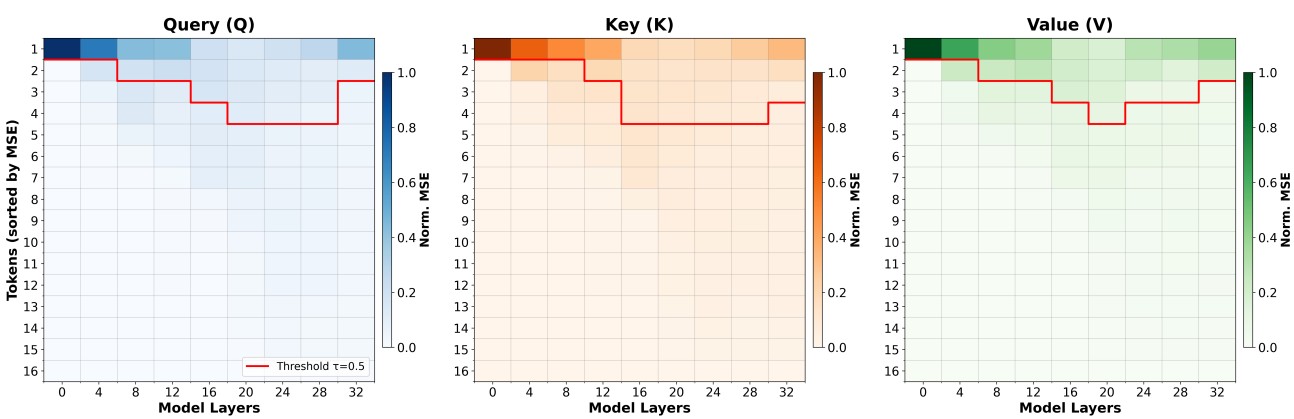

*Figure 9.* Combined QKV Normalized MSE Heatmap with Threshold Selection ($\tau = 0.5$) for Data Sample 100.

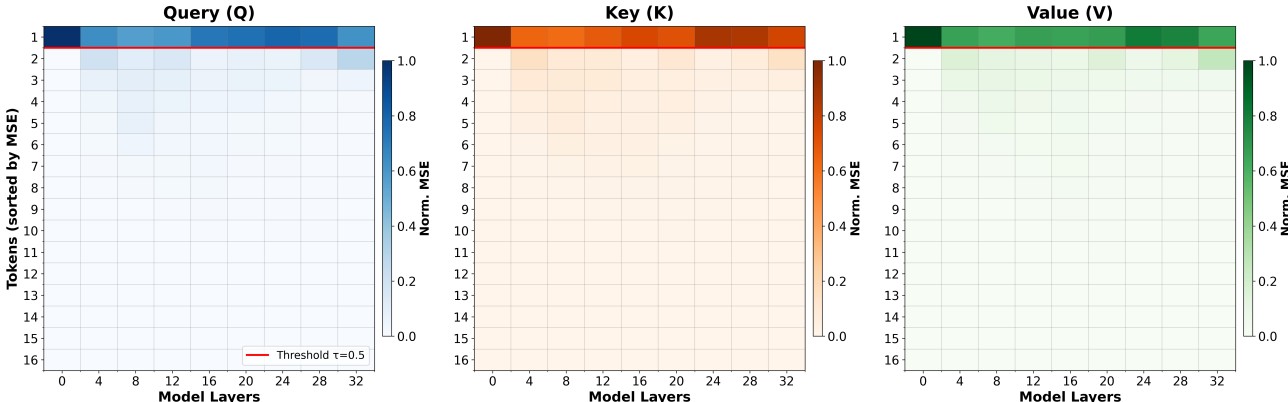

*Figure 10.* Combined QKV Normalized MSE Heatmap with Threshold Selection ($\tau = 0.5$) for Data Sample 200.

*Figure 11.* Combined QKV Normalized MSE Heatmap with Threshold Selection ($\tau = 0.5$) for Data Sample 300.

