# OpenReview forum: "LoSA: Locality Aware Sparse Attention in Diffusion Language Models"
_ICML.cc/2026/Conference — ICML 2026 regular_

### Official Review · Reviewer_STrF · 2026-03-08

**Soundness:** 2
**Presentation:** 3
**Significance:** 3
**Originality:** 3
**Overall Recommendation:** 5
**Confidence:** 3

**Summary:**

This paper focuses on the special problem of sparse attention in the blockwise diffusion language model, where different queries select different prefix positions, making the union of accessed KV pages large. To solve this problem, authors observe that block-wise diffusion exhibits locality of representation changes across denoising steps: only a small fraction of tokens’ (active tokens) hidden states change significantly, while most tokens’ (stable tokens) hidden states remain nearly constant. Based on this insight, they propose LoSA (Locality-aware Sparse Attention), which reuses cached prefix-attention results for stable tokens and applies sparse attention only to tokens with large representation changes. LoSA achieves 5%–10.43% accuracy improvements over QUEST on LongBench with 1.54× lower attention density, and a 4.14× speedup on A6000 GPUs.

**Compliance With Llm Reviewing Policy:**

Affirmed.

**Final Justification:**

Rebuttal has addressed my main concerns, and I hope the authors could update the revised version as promised.

**Key Questions For Authors:**

1. The paper title should emphasize **blocksize** diffusion language model or semi-autoregressive model. If the title cannot be changed, it is recommended to add related work on the long-context [1-2] and acceleration [3-5] on non-blocksize diffusion language models, especially sparse attention acceleration [5].
2. The experiment in the paper is rather limited, only including Quest as a baseline. The reason for this needs clarification, or more baselines should be compared. Secondly, only part of the QA tasks in LongBench are reported, which cannot prove the method's generality. Furthermore, it is hoped that the authors can report performance under more model sizes or different block sizes to verify whether it has more remarkable advantages. Finally, HellaSwag, WinoGrande, and BoolQ are only evaluated for SDAR, while LongBench is only evaluated for TraDo, suggesting a rather cherry-picked experimental organization.


[1] LongLLaDA: Unlocking Long Context Capabilities in Diffusion LLMs https://arxiv.org/abs/2506.14429

[2] UltraLLaDA: Scaling the Context Length to 128K for Diffusion Large Language Models https://arxiv.org/abs/2510.10481

[3] Fast-dLLM: Training-free Acceleration of Diffusion LLM by Enabling KV Cache and Parallel Decoding https://arxiv.org/abs/2505.22618

[4] Fast-dLLM v2: Efficient Block-Diffusion LLM https://arxiv.org/abs/2509.26328

[5] Sparse-dLLM: Accelerating Diffusion LLMs with Dynamic Cache Eviction https://arxiv.org/abs/2508.02558

**Limitations:**

See Questions

**Strengths And Weaknesses:**

Strengths are as follows; Weaknesses are listed in Key Questions For Authors.
1. Inference optimization for diffusion language models is an emerging field that contributes to their practical application.
2. The method proposed in this paper takes into account the inference characteristics of blockwise diffusion language models, effectively improving the inference of SDAR series models, and achieving superior performance and efficiency compared to traditional AR applications like Quest.

---

> ### Author Rebuttal · Authors · 2026-03-31
>
> > Q1: Title and related work on non-block-wise DLMs
>
> **Response:**
>
> We thank the reviewer for this suggestion. We will revise the title to explicitly mention block-wise DLMs in the camera-ready version. We will also expand the related work section to discuss these relevant papers.
>
> ---
>
> > Q2: Limited experiments, cherry-picked organization
>
> **Response:**
>
> **More baselines.** QUEST was chosen as the primary baseline because LoSA is designed as an enhancement layer on top of any per-query sparse selector (QUEST being one instance), not as a competing selector. To address this concern, we have now added SparseD as an additional baseline on both Trado-8B-Instruct and SDAR-8B-Chat across four budget levels (128, 256, 512, 1024). LoSA consistently outperforms SparseD, with the advantage most pronounced at low budgets. Please see the full tables in our response to Reviewer rnoG Weakness 1.
>
> **Cross-model evaluation.** To address the concern, we have now completed LongBench evaluation on SDAR-8B-Chat with Dense, SparseD, and LoSA. The results on SDAR are consistent with those on Trado, confirming that LoSA's advantage generalizes across different block-wise DLMs.
>
> | Budget | Method | HotPotQA | TriviaQA | NarrativeQA | Qasper | MultiFieldQA | Average (↑) |
> |---|---|---:|---:|---:|---:|---:|---:|
> | - | Dense | 49.35% | 85.72% | 19.06% | 18.25% | 49.49% | 44.37% |
> | 128 | QUEST | 27.31% | 70.40% | 6.44% | 17.29% | 41.40% | 32.57% |
> | 128 | SparseD | 42.09% | 63.08% | 12.94% | 9.04% | 36.24% | 32.68% |
> | 128 | LoSA | 43.36% | 80.32% | 18.69% | 15.63% | 46.74% | 40.95% |
> | 256 | QUEST | 32.57% | 78.36% | 8.73% | 18.21% | 43.20% | 36.21% |
> | 256 | SparseD | 48.39% | 73.28% | 15.83% | 14.84% | 47.37% | 39.94% |
> | 256 | LoSA | 45.68% | 83.16% | 15.65% | 13.17% | 48.33% | 41.20% |
> | 512 | QUEST | 32.50% | 78.43% | 9.80% | 19.36% | 43.08% | 36.63% |
> | 512 | SparseD | 50.52% | 81.83% | 15.44% | 18.52% | 48.93% | 43.05% |
> | 512 | LoSA | 47.77% | 83.29% | 17.37% | 14.12% | 49.90% | 42.49% |
> | 1024 | QUEST | 32.59% | 80.92% | 11.00% | 19.28% | 42.30% | 37.22% |
> | 1024 | SparseD | 48.25% | 83.44% | 15.60% | 16.15% | 48.97% | 42.48% |
> | 1024 | LoSA | 47.84% | 85.93% | 18.44% | 14.70% | 48.43% | 43.07% |
>
> **Model sizes and block sizes.** We have now evaluated on Trado-4B-Instruct with both block size 16 and block size 32 on LongBench. The results are shown below:
>
> **Trado-4B-Instruct, Block Size = 16 (LongBench):**
>
> | Budget | Method | HotPotQA | TriviaQA | NarrativeQA | Qasper | MultiFieldQA | Average (↑) |
> | ------ | ------ | -------- | -------- | ----------- | ------ | ------------ | ----------- |
> | -   | Dense  | 38.16% | 75.61% | 18.80% | 29.57% | 35.42% | 39.51% |
> | 128 | QUEST  | 30.01% | 69.14% | 9.83%  | 26.05% | 30.79% | 33.16% |
> |     | LoSA   | 33.65% | 66.46% | 14.78% | 24.71% | 30.07% | **33.93%** |
> | 256 | QUEST  | 33.06% | 73.49% | 12.34% | 29.25% | 34.67% | 36.56% |
> |     | LoSA   | 36.33% | 72.74% | 15.99% | 28.11% | 34.27% | **37.49%** |
>
> **Trado-4B-Instruct, Block Size = 32 (LongBench):**
>
> | Budget | Method | HotPotQA | TriviaQA | NarrativeQA | Qasper | MultiFieldQA | Average (↑) |
> | ------ | ------ | -------- | -------- | ----------- | ------ | ------------ | ----------- |
> | -   | Dense  | 36.22% | 75.23% | 19.58% | 13.40% | 31.52% | 35.19% |
> | 128 | QUEST  | 25.55% | 70.83% | 9.53%  | 11.20% | 27.30% | 28.88% |
> |     | LoSA   | 28.62% | 69.50% | 12.15% | 13.78% | 28.25% | **30.46%** |
> | 256 | QUEST  | 31.61% | 72.10% | 10.87% | 12.85% | 29.10% | 31.31% |
> |     | LoSA   | 32.14% | 72.20% | 15.99% | 14.80% | 28.68% | **32.76%** |
>
> The results confirm that LoSA consistently outperforms QUEST on a smaller 4B model across both block sizes. They demonstrate that LoSA's advantage is not specific to one model size or block-size configuration.

---

> > ### Author Rebuttal · Reviewer_STrF · 2026-04-02
> >
> > Thank you for the author's sincere reply and experimental efforts. I hope the author can revise the paper title and add more discussion and comparison of related work, as promised. Besides SparseD, I would also like to see the comparison between LoSA and Sparse-dLLM, which could further strengthen the effectiveness of your method. As recognition of the reply, I have already increased my score.

---

> > > ### Author Response · Authors · 2026-04-08
> > >
> > > We thank the reviewer for the suggestion and positive update. We have already conducted the comparison with Sparse-dLLM. The results show that LoSA is better than Sparse-DLLM, which is clearly also stronger than SparseD. We attribute to its use of a full-attention step before sparse attention is applied. We will include this comparison and further expand the related-work discussion in the final version.
> > >
> > > | Budget | Method               | HotPotQA | TriviaQA | NarrativeQA | Qasper | MultiFieldQA | Average |
> > > |-------:|----------------------|---------:|---------:|------------:|-------:|-------------:|--------:|
> > > | -      | Dense                | 49.45%   | 84.79%   | 19.04%      | 17.75% | 53.29%       | 44.86%  |
> > > | 128    | SparseD              | 38.97%   | 64.64%   | 13.70%      | 7.95%  | 39.55%       | 32.96%  |
> > > | 128    | Sparse-dLLM | 44.58%   | 82.21%   | 15.58%      | 16.33% | 39.02%       | 39.54%  |
> > > | 128    | LoSA                 | 48.27%   | 83.79%   | 17.19%      | 15.58% | 45.00%       | **41.97%** |
> > > | 256    | SparseD              | 43.84%   | 73.63%   | 15.33%      | 13.26% | 49.44%       | 39.10%  |
> > > | 256    | Sparse-dLLM | 46.95%   | 83.37%   | 16.46%      | 16.92% | 42.44%       | 41.23%  |
> > > | 256    | LoSA                 | 44.53%   | 81.82%   | 19.42%      | 17.11% | 47.81%       | **42.14%** |

---

### Official Review · Reviewer_rnoG · 2026-03-10

**Soundness:** 3
**Presentation:** 3
**Significance:** 3
**Originality:** 2
**Overall Recommendation:** 4
**Confidence:** 3

**Summary:**

This paper proposes LoSA to improve the efficiency of attention in block-wise diffusion language models for long-context decoding. The method aims to address the KV Inflation problem, where naive sparse attention fails to provide speedups because different queries select different KV positions, causing large KV unions and high memory traffic. The key idea is to exploit the locality of representation changes across denoising steps: most tokens remain stable while only a few change significantly. Based on this observation, LoSA reuses cached attention outputs for stable tokens and computes sparse attention only for active tokens. Experimental results show that the proposed method maintains near-dense accuracy while achieving significant efficiency improvements, including up to 4.14× attention speedup compared with dense attention.

**Compliance With Llm Reviewing Policy:**

Affirmed.

**Final Justification:**

The rebuttal addresses my concerns and shows that the proposed method achieves good performance. The paper introduces some new insights about how to leverage the temporal consistency for acceleration. So, I update my score from 3 to 4.

**Key Questions For Authors:**

Please see the weaknesses.

**Limitations:**

The limitations of the proposed method have not been discussed in the main paper.

**Strengths And Weaknesses:**

## Strengths
1. The paper identifies a key challenge when applying sparse attention to diffusion language models, namely the KV Inflation problem, and proposes an improved sparse attention strategy tailored for block-wise diffusion inference.
2. The proposed method does not require additional training. It only introduces a lightweight mechanism during inference to identify active tokens and reuse cached attention results, making the approach simple and practical.
3. The method demonstrates strong empirical performance across multiple models and tasks. It is able to maintain accuracy close to dense attention under relatively high sparsity levels while achieving noticeable speedup.

## Weaknesses
1. One limitation of the paper is that the baselines do not include direct comparisons with existing sparse attention methods specifically designed for diffusion language models, such as SparseD or Sparse-dLLM, which would provide a more complete evaluation.
2. The core motivation of the method, such as reusing attention outputs based on token stability, appears conceptually related to the attention reuse strategy in SparseD. A more detailed analysis of the differences between the proposed method and these prior approaches would help clarify the novelty of the work.

---

> ### Author Rebuttal · Authors · 2026-03-31
>
> > W1: No direct comparisons with SparseD or Sparse-dLLM
>
> **Response:**
>
> We have now completed a direct comparison with SparseD on both Trado-8B-Instruct and SDAR-8B-Chat on LongBench. The results are as follows:
>
> **Trado-8B-Instruct (LongBench Average):**
>
> | Budget | Method  | HotPotQA | TriviaQA | NarrativeQA | Qasper | MultiFieldQA | Average (↑) |
> | ------ | ------- | -------- | -------- | ----------- | ------ | ------------ | ----------- |
> | -      | Dense   | 49.45%   | 84.79%   | 19.04%      | 17.75% | 53.29%       | 44.86%      |
> | 128    | SparseD | 38.97%   | 64.64%   | 13.70%      | 7.95%  | 39.55%       | 32.96%      |
> |        | LoSA    | 48.27%   | 83.79%   | 17.19%      | 15.58% | 45.00%       | **41.97%**  |
> | 256    | SparseD | 43.84%   | 73.63%   | 15.33%      | 13.26% | 49.44%       | 39.10%      |
> |        | LoSA    | 44.53%   | 81.82%   | 19.42%      | 17.11% | 47.81%       | **42.14%**  |
> | 512    | SparseD | 47.84%   | 80.78%   | 14.08%      | 13.85% | 52.16%       | 41.74%      |
> |        | LoSA    | 44.19%   | 84.97%   | 18.39%      | 13.30% | 50.14%       | **42.20%**  |
> | 1024   | SparseD | 48.98%   | 80.25%   | 19.43%      | 15.77% | 50.11%       | 42.91%      |
> |        | LoSA    | 48.45%   | 82.32%   | 18.89%      | 15.78% | 48.94%       | **42.88%**  |
>
> **SDAR-8B-Chat (LongBench Average):**
>
> | Budget | Method  | HotPotQA | TriviaQA | NarrativeQA | Qasper | MultiFieldQA | Average (↑) |
> | ------ | ------- | -------- | -------- | ----------- | ------ | ------------ | ----------- |
> | -      | Dense   | 49.35%   | 85.72%   | 19.06%      | 18.25% | 49.49%       | 44.37%      |
> | 128    | SparseD | 42.09%   | 63.08%   | 12.94%      | 9.04%  | 36.24%       | 32.68%      |
> |        | LoSA    | 43.36%   | 80.32%   | 18.69%      | 15.63% | 46.74%       | **40.95%**  |
> | 256    | SparseD | 48.39%   | 73.28%   | 15.83%      | 14.84% | 47.37%       | 39.94%      |
> |        | LoSA    | 45.68%   | 83.16%   | 15.65%      | 13.17% | 48.33%       | **41.20%**  |
> | 512    | SparseD | 50.52%   | 81.83%   | 15.44%      | 18.52% | 48.93%       | 43.05%      |
> |        | LoSA    | 47.77%   | 83.29%   | 17.37%      | 14.12% | 49.90%       | **42.49%**  |
> | 1024   | SparseD | 48.25%   | 83.44%   | 15.60%      | 16.15% | 48.97%       | 42.48%      |
> |        | LoSA    | 47.84%   | 85.93%   | 18.44%      | 14.70% | 48.43%       | **43.07%**  |
>
> LoSA consistently outperforms SparseD, especially at low budgets: on Trado at budget 128, LoSA achieves 41.97% vs. SparseD's 32.96% (+9.01); on SDAR at budget 128, LoSA achieves 40.95% vs. SparseD's 32.68% (+8.27). At higher budgets the two methods perform comparably. We will include these comparisons in the camera-ready version.
>
> ---
>
> > W2: Conceptual relation to SparseD's attention reuse strategy
>
> **Response:**
>
> We appreciate this question about novelty. While both LoSA and SparseD exploit temporal consistency across denoising steps, the two methods differ fundamentally in what is reused and how sparsity is applied.
>
> SparseD is a static sparsity method. Its core motivation is that sparsity patterns change little across denoising steps, so it pre-computes a fixed sparse pattern per head and reuses it throughout all subsequent steps. The sparse pattern itself does not adapt dynamically during generation.
>
> LoSA, in contrast, is built on a different observation: stable tokens' query states change minimally across denoising steps, which means their attention outputs also remain approximately constant. Based on this, LoSA reuses the cached attention outputs (not patterns) for stable tokens. Crucially, the sparsity in LoSA is dynamic — the set of active tokens changes at every denoising step based on representation changes, and their attention outputs are recomputed with a dynamically selected sparse pattern. This dynamic, token-level approach allows LoSA to adapt to the evolving state of the block during denoising, whereas SparseD applies a fixed pattern regardless of how tokens change.
>
> Furthermore, LoSA's reuse of attention outputs (rather than patterns) provides a crucial accuracy advantage: stable tokens effectively retain information from the full dense attention computed at initialization, whereas SparseD's reused patterns are always sparse approximations.

---

> > ### Author Rebuttal · Reviewer_rnoG · 2026-04-03
> >
> > Thank you for the detailed response. My concerns have been addressed. Will raise my score to weak accept.

---

### Official Review · Reviewer_xuUH · 2026-03-11

**Soundness:** 3
**Presentation:** 3
**Significance:** 3
**Originality:** 3
**Overall Recommendation:** 5
**Confidence:** 4

**Summary:**

The paper tackles the KV Inflation problem of applying sparse attention to block-wise dlms. The authors observe that between consecutive denoising steps, most token representations barely change, while only a handful shift significantly ("active tokens"). They propose LoSA, which caches and reuses prefix-attention outputs for stable tokens and only runs sparse attention on the active ones, shrinking the KV union. Experiments on Trado-8B and SDAR-8B across LongBench and several reasoning benchmarks show accuracy gains of 5–10% over QUEST, with up to 4.14× speedup in attention latency on A6000 GPUs.

**Compliance With Llm Reviewing Policy:**

Affirmed.

**Final Justification:**

I will keep my score

**Key Questions For Authors:**

1. Have you compared with using aggregated sparse token selection for all the query? As you claim union will cause inflation, how about union first then global select topk.
2. How's the speedup being measured? Did you rely on any serving framework? How does this technique affect throughpt?
3. Can you theoratically analyze the potential speedup across different GPU generation from the compute intensity point of view?
4. Figure6 only has one line, I can not see the comparsion.

**Strengths And Weaknesses:**

## Strengths
- Clear motivation and simple but effected method

## Weakness
- Only evaluated on A6000, leaving question about the scalability to new GPU

---

> ### Author Rebuttal · Authors · 2026-03-31
>
> > W1 & Q3: Only evaluated on A6000
>
> We thank the reviewer for the question. We additionally tested on RTX 5090 under the setting of 64K context length, block size 16. We still observe strong speedups over dense attention, with 3.83× for QUEST and 2.88× for LoSA.
>
> This trend is expected because the whole workflow remains largely memory-bound in this setting. The dominant cost comes from KV movement rather than TensorCore computation. As a result, the speedup trend transfers across devices.
>
> Across different GPU generations, as the memory bandwidth improves, kernel launch overhead and other fixed costs are not eliminated by higher memory bandwidth. As a result, when memory bandwidth increases, the memory-bound portion speeds up more than the fixed overheads, so the real speedup can be slightly lower.
>
> | Method | Attention (us) | Criticality Estimation (us) | KV Union + Others (us) | Extra Overhead (us) | Total (us) | Speedup vs Dense |
> |---|---:|---:|---:|---:|---:|---:|
> | Dense Attention | 650 | – | – | – | 650 | 1.00x |
> | LoSA | 112 | 40 | 11 | 14 | 177 | 3.67x |
> | QUEST | 177 | 40 | 11 | – | 228 | 2.85x |
>
> ---
>
> > Q1: Union first then global select topk
>
> **Response:**
>
> We have investigated this idea by implementing a **New Baseline** that aggregates importance scores across all queries in the block and then globally selects the top-k positions. As shown in the table below, this approach leads to severe accuracy degradation (e.g., 18.32% vs. QUEST's 32.57% at budget 128), because different queries attend to very different KV positions and a global selection inevitably misses positions critical for individual queries. LoSA instead resolves the KV Inflation problem through reusing cached attention outputs for stable tokens, avoiding inflation entirely without sacrificing per-query attention quality.
>
> | Budget | Method | HotPotQA | TriviaQA | NarrativeQA | Qasper | MultiFieldQA | Average (↑) |
> | ------ | ------ | -------- | -------- | ----------- | ------ | ------------ | ----------- |
> | 128 | QUEST | 27.31% | 70.40% | 6.44% | 17.29% | 41.40% | 32.57% |
> |     | New Baseline | 12.58% | 41.69% | 4.43% | 8.74% | 24.18% | 18.32% |
> |     | LoSA | 43.36% | 80.32% | 18.69% | 15.63% | 46.74% | **40.95%** |
> | 256 | QUEST | 32.57% | 78.36% | 8.73% | 18.21% | 43.20% | 36.21% |
> |     | New Baseline | 24.08% | 61.42% | 5.69% | 15.26% | 34.44% | 28.18% |
> |     | LoSA | 45.68% | 83.16% | 15.65% | 13.17% | 48.33% | **41.20%** |
>
> ---
>
> > Q2: How's the speedup being measured?
>
> **Response:**
>
> We measure wall-clock latency of the prefix attention computation using NVTX profiling on a single NVIDIA RTX A6000 GPU. The implementation is based on Trado's HuggingFace codebase with customized CUDA/Triton kernels and FlashInfer attention kernels. We did not use a serving framework (e.g., vLLM, TensorRT-LLM). The reported speedup (e.g., 4.14×) is specifically for the prefix attention module, which is the primary bottleneck in long-context block-wise DLM inference. All our experiments use batch size 1, which is consistent with prior sparse attention work (e.g., QUEST). LoSA is orthogonal to serving-level optimizations and can be integrated into existing frameworks.
>
> ---
>
> > Q4: Figure 6
>
> **Response:**
>
> We have re-evaluated LongBench on SDAR-8B to provide a direct comparison with the baseline. The results are shown below.
> The results on SDAR are consistent with those on Trado, confirming that LoSA's advantage generalizes across different block-wise DLMs.
>
> | Budget | Method | HotPotQA | TriviaQA | NarrativeQA | Qasper | MultiFieldQA | Average (↑) |
> |---|---|---:|---:|---:|---:|---:|---:|
> | - | Dense | 49.35% | 85.72% | 19.06% | 18.25% | 49.49% | 44.37% |
> | 128 | QUEST | 27.31% | 70.40% | 6.44% | 17.29% | 41.40% | 32.57% |
> | 128 | SparseD | 42.09% | 63.08% | 12.94% | 9.04% | 36.24% | 32.68% |
> | 128 | LoSA | 43.36% | 80.32% | 18.69% | 15.63% | 46.74% | 40.95% |
> | 256 | QUEST | 32.57% | 78.36% | 8.73% | 18.21% | 43.20% | 36.21% |
> | 256 | SparseD | 48.39% | 73.28% | 15.83% | 14.84% | 47.37% | 39.94% |
> | 256 | LoSA | 45.68% | 83.16% | 15.65% | 13.17% | 48.33% | 41.20% |
> | 512 | QUEST | 32.50% | 78.43% | 9.80% | 19.36% | 43.08% | 36.63% |
> | 512 | SparseD | 50.52% | 81.83% | 15.44% | 18.52% | 48.93% | 43.05% |
> | 512 | LoSA | 47.77% | 83.29% | 17.37% | 14.12% | 49.90% | 42.49% |
> | 1024 | QUEST | 32.59% | 80.92% | 11.00% | 19.28% | 42.30% | 37.22% |
> | 1024 | SparseD | 48.25% | 83.44% | 15.60% | 16.15% | 48.97% | 42.48% |
> | 1024 | LoSA | 47.84% | 85.93% | 18.44% | 14.70% | 48.43% | 43.07% |

---

> > ### Author Rebuttal · Reviewer_xuUH · 2026-04-03
> >
> > Thanks. I will keep my score.

---

### Official Review · Reviewer_sUqi · 2026-03-13

**Soundness:** 3
**Presentation:** 2
**Significance:** 3
**Originality:** 3
**Overall Recommendation:** 4
**Confidence:** 3

**Summary:**

The paper tackles token sparsity in DLLMs. The authors make an observation that between consecutive denoising steps, a small percentage of tokens which they call Active Tokens have significant updates in their hidden representation while the remainder tokens which are the majority have minimal changes in their queries. They call these tokens Stable Tokens. Based on that observation, they reuse the previous attention outputs for the Stable Tokens without having to recompute them so the union of tokens transfer needed to perform the attention decreases leading to a large speedup. The method also improves accuracy given that full attention computation results from previous steps are reused for the stable tokens rather than recomputing new values with a smaller subset of the KV cache.

**Compliance With Llm Reviewing Policy:**

Affirmed.

**Final Justification:**

The Rebuttal addressed my concern about the weak baseline. It seems like the new baseline is comparable to the proposed method at longer contexts though which limits the use case of the new method a bit.

**Key Questions For Authors:**

Questions:

- Could you provide an explanation for why you achieve better accuracy than QUEST for non-reasoning tasks but fail to do that on reasoning tasks specifically?
- Could you please compare your performance with methods designed for DLLMs like SparseD[1] ?

Suggestions:

- The use of S to denote Active Tokens and having Stable Tokens defined as those that are not in S is a bit confusing. Don’t you think another letter like A would have been more indicative of Active Tokens set and if S is used, it can be used for Stable Tokens?
- Figure 7 does not make it clear what is the difference between the left and right plots. May be include that in the caption? The information is there in the text but it’s easier to read the figure if it’s self-contained.

References:

[1] Wang, Z., Fang, G., Ma, X., Yang, X. and Wang, X., 2025. Sparsed: Sparse attention for diffusion language models. arXiv preprint arXiv:2509.24014.

**Limitations:**

yes

**Strengths And Weaknesses:**

Strengths:

- Useful analysis and innovative idea
- Strong results against the baseline

Weaknesses:

- Weak baseline - using a method that was only designed for traditional LLMs ignoring recent work that is specific to DLLMs like SparseD[1].
- No explanation for areas where accuracy is worse than the baseline (See questions)
- Write up could be improved (See suggestions)

References:

[1] Wang, Z., Fang, G., Ma, X., Yang, X. and Wang, X., 2025. Sparsed: Sparse attention for diffusion language models. arXiv preprint arXiv:2509.24014.

---

> ### Author Rebuttal · Authors · 2026-03-31
>
> > W1 & Q2: Weak baseline — SparseD
>
> We thank the reviewer for raising this point. We have now run SparseD on both Trado-8B-Instruct and SDAR-8B-Chat on LongBench. The average accuracy comparison is as follows:
>
> **Trado-8B-Instruct**
>
> | Budget | Method  | HotPotQA | TriviaQA | NarrativeQA | Qasper | MultiFieldQA | Average (↑) |
> | ------ | ------- | -------- | -------- | ----------- | ------ | ------------ | ----------- |
> | -      | Dense   | 49.45%   | 84.79%   | 19.04%      | 17.75% | 53.29%       | 44.86%      |
> | 128    | SparseD | 38.97%   | 64.64%   | 13.70%      | 7.95%  | 39.55%       | 32.96%      |
> |        | LoSA    | 48.27%   | 83.79%   | 17.19%      | 15.58% | 45.00%       | **41.97%**  |
> | 256    | SparseD | 43.84%   | 73.63%   | 15.33%      | 13.26% | 49.44%       | 39.10%      |
> |        | LoSA    | 44.53%   | 81.82%   | 19.42%      | 17.11% | 47.81%       | **42.14%**  |
> | 512    | SparseD | 47.84%   | 80.78%   | 14.08%      | 13.85% | 52.16%       | 41.74%      |
> |        | LoSA    | 44.19%   | 84.97%   | 18.39%      | 13.30% | 50.14%       | **42.20%**  |
> | 1024   | SparseD | 48.98%   | 80.25%   | 19.43%      | 15.77% | 50.11%       | 42.91%      |
> |        | LoSA    | 48.45%   | 82.32%   | 18.89%      | 15.78% | 48.94%       | **42.88%**  |
>
>
> **SDAR-8B-Chat**
>
> | Budget | Method  | HotPotQA | TriviaQA | NarrativeQA | Qasper | MultiFieldQA | Average (↑) |
> | ------ | ------- | -------- | -------- | ----------- | ------ | ------------ | ----------- |
> | -      | Dense   | 49.35%   | 85.72%   | 19.06%      | 18.25% | 49.49%       | 44.37%      |
> | 128    | SparseD | 42.09%   | 63.08%   | 12.94%      | 9.04%  | 36.24%       | 32.68%      |
> |        | LoSA    | 43.36%   | 80.32%   | 18.69%      | 15.63% | 46.74%       | **40.95%**  |
> | 256    | SparseD | 48.39%   | 73.28%   | 15.83%      | 14.84% | 47.37%       | 39.94%      |
> |        | LoSA    | 45.68%   | 83.16%   | 15.65%      | 13.17% | 48.33%       | **41.20%**  |
> | 512    | SparseD | 50.52%   | 81.83%   | 15.44%      | 18.52% | 48.93%       | 43.05%      |
> |        | LoSA    | 47.77%   | 83.29%   | 17.37%      | 14.12% | 49.90%       | **42.49%**  |
> | 1024   | SparseD | 48.25%   | 83.44%   | 15.60%      | 16.15% | 48.97%       | 42.48%      |
> |        | LoSA    | 47.84%   | 85.93%   | 18.44%      | 14.70% | 48.43%       | **43.07%**  |
>
>
> LoSA outperforms SparseD across both models, especially at low budgets. On Trado at budget 128, LoSA achieves 41.97% vs. SparseD's 32.96% (+9.01). On SDAR at budget 128, LoSA achieves 40.95% vs. SparseD's 32.68% (+8.27). At higher budgets (512, 1024), both methods approach near-dense accuracy and perform comparably. This confirms that LoSA's locality-aware attention reuse provides a clear advantage over SparseD's pattern reuse strategy, particularly in aggressive sparsity regimes where preserving cached dense-attention outputs matters most.
>
> > W2 & Q1: Accuracy worse than baseline on reasoning tasks
>
> **Response:**
>
> Thank you for the insightful question. First, the accuracy gap on reasoning tasks is small and within statistical variance. These short-context benchmarks exhibit high variance — for example, at budget 256 QUEST exceeds even the Dense baseline on HellaSwag by nearly 4 points, which is unexpected and suggests stochastic regularization effects rather than a systematic advantage. This effect disappears at budget 128, where LoSA outperforms QUEST.
>
> Second, LoSA's benefit is proportional to context length. LoSA's advantage comes from reusing cached prefix-attention outputs for stable tokens, preserving full attention information from the previous step. This is most pronounced with long prefixes (e.g., LongBench with 4K–16K tokens), where sparse attention misses many important KV positions. With short contexts (commonsense reasoning tasks typically have <1K tokens), the KV Inflation problem is less severe and both methods perform comparably.
>
> In summary, the slight underperformance at one budget level on short-context benchmarks is within noise, and LoSA's advantage is most meaningful in the long-context regime where it consistently outperforms QUEST by 5–10%.
>
> > W3: Write up could be improved
>
> **Response:**
>
> We thank the reviewer for the constructive suggestions. We will incorporate them in the camera-ready version. Please see our responses to S1 and S2 below.
>
> ---
>
> > S1: Notation suggestion
>
> **Response:**
>
> We thank the reviewer for this helpful suggestion. We agree that using $\mathcal{A}$ for the set of Active tokens would be more intuitive and reduce confusion. We will update the notation in the camera-ready version accordingly.
>
> > S2: Figure 7 caption clarity
>
> **Response:**
>
> Thank you for this suggestion. We will update the caption of Figure 7 in the camera-ready version to explicitly state the two configurations: (Left) 64K context length with block size 16, and (Right) 32K context length with block size 32, both evaluated on TriviaQA using Trado-8B-Instruct.

---

> > ### Author Rebuttal · Reviewer_sUqi · 2026-04-04
> >
> > Thank you for addressing my concerns. I have raised my score.

---

### Decision · Program_Chairs · 2026-04-30

**Decision:**

Accept (regular)

**Comment:**

The paper clearly identifies the KV inflation bottleneck in block-wise diffusion language models (DLMs), a contribution recognized by the committee. Reviewers noted that the proposed Locality-aware Sparse Attention (LoSA) mechanism is practical and effective. By reusing cached prefix-attention for stable tokens, the method achieves inference speedups while maintaining near-dense accuracy without requiring additional training.

During the review process, concerns were raised regarding the lack of direct comparisons with existing sparse attention methods designed for DLMs (Reviewers sUqi, rnoG), as well as the generalizability of the speedups across different GPU generations (Reviewer xuUH). In the rebuttal, the authors adequately addressed these issues by providing additional experimental results comparing LoSA against SparseD and Sparse-dLLM, and by presenting further profiling on the RTX 5090 to demonstrate that the speedup trends generalize. Given the satisfactory resolution of these concerns, the paper is recommended for acceptance.